# Deposition of complement regulators on the surface of *Plasmodium falciparum* merozoites depends on the immune status of the host

Maria Rosaria Bassi[1☯], Bogdan Cristinoi[1☯], Frank Buitenwerf[1], Mark Bergholt Cuadrado[1], Kasper Haldrup Björnsson[1], Melanie Rose Walker[1], Frederica Dedo Partey[2], Andrew B. Ward[3], Michael Fokuo Ofori[2], Lea Barfod[1]*

1 Centre for Translational Medicine and Parasitology at Department of Immunology and Microbiology (ISIM), Faculty of Health and Medical Sciences, University of Copenhagen, Copenhagen, Denmark, 2 Noguchi Memorial Institute for Medical Research, University of Ghana, Legon, Ghana, 3 Department of Integrative Structural and Computational Biology, The Scripps Research Institute, La Jolla, California, United States of America

☯ These authors contributed equally to the work.

* lbarfod@sund.ku.dk

## Abstract

*Plasmodium falciparum* is responsible for the majority of malaria cases and deaths worldwide. In malaria endemic areas, natural immunity to blood stage infection is acquired over several exposures to the parasite and is thought to rely on antibodies. Antibodies can protect from severe disease through different effector functions, with complement activation lately emerging as an important feature of protective humoral responses to malaria. Plasmodium parasites have however evolved several mechanisms to evade complement attack, including the recruitment of complement down-regulatory proteins like Factor H (FH) and C1 esterase inhibitor (C1-INH). In this study, we report that merozoite-specific antibodies acquired naturally after infection activate the complement cascade in an exposure-dependent manner. Using plasma samples from convalescent children and exposed adults collected respectively in Hohoe and Accra (Ghana), we show that the ability to fix C1q and activate the classical pathway is similar for antibodies deriving from the two donors groups. However, downstream complement activation shown as deposition of the membrane attack complex (MAC) is strikingly higher with antibodies from children compared to antibodies from adults. Moreover, we demonstrate that antibodies from naturally exposed children can interfere with the merozoite recruitment of FH, but not of C1-INH. With the aim of neutralizing parasite evasion of the complement classical pathway, we develop a murine monoclonal antibody targeting PfMSP3, the binding partner of C1-INH on the merozoite surface. We demonstrate that this antibody can effectively block the binding of C1-INH to the parasite surface, unlike the naturally acquired ones. Using cryogenic electron microscopy, we obtain a low-resolution structure of the monoclonal antibody in complex with

**Data availability statement:** The electron density map presented in this paper is available on the Electron Microscopy Data Bank (EMDB) under accession code EMD-51634. All the other data are within the manuscript and its Supporting Information files.

**Funding:** This study was funded by Novo Nordisk Fonden, grant number NNF170C0026778 awarded to L.B. and by the EliteForsk Travelling Grant, grant number: 3142-00038B awarded by the Danish Ministry of Higher Education and Science to K.B. The funders had no role in study design, data collection and analysis, decision to publish, or preparation of the manuscript.

**Competing interests:** The authors have declared that no competing interests exist

PfMSP3, which is the first reported structural data for this antigen. We propose targeting parasite antigens binding to complement down-regulators, together with leading vaccine candidate antigens, as a novel strategy to enhance the efficacy of future malaria vaccines.

---

## Author summary

The immune mechanisms mediating clinical immunity to malaria are still not fully understood, but antibodies are considered the main defense towards the symptomatic blood stage of the disease. Recent studies have highlighted the importance of complement activation in malaria immunity, however malaria parasites are capable of evading complement recognition. In this study we provide novel insights on how the complement system is activated or evaded in the presence of malaria-specific antibodies developed after natural infection. We show that antibodies from convalescent children can inhibit parasite growth and neutralize complement evasion more effectively than antibodies from immune adults. Moreover, we demonstrate that vaccination with a malaria antigen involved in complement down regulation can induce antibodies that can restore complement activation state better than naturally acquired antibodies. We suggest including such antigens in future malaria vaccines to obtain superior antibody efficacy through the activation of the complement system. These findings can contribute to advance malaria vaccine development.

## Introduction

Nearly half of the world´s population is at risk of malaria and in 2022, there were estimated 249 million malaria cases globally [1]. *Plasmodium* parasites have a complex life cycle involving different stages of replication. During the symptomatic blood stage of the disease, merozoites invade red blood cells (RBCs), in which they divide and eventually rupture, releasing new merozoites that continue the infection cycle [2]. Naturally acquired immunity to malaria is non-sterile and develops slowly over time after repeated exposures to the pathogen [3,4]. Humoral immunity is crucial in the host response to Plasmodium infection, particularly during the blood stage, when merozoites are released by bursting infected erythrocytes, making the parasite vulnerable to antibody-mediated neutralization [5,6].

Several growth inhibitory monoclonal and polyclonal antibodies targeting merozoite antigens have been described so far [7–12]. In particular, antibodies specific for the PfPTRAMP–PfCSS–PfRipr–CyRPA–PfRh5 (PCRCR) invasion complex have shown the ability of blocking merozoite entry into RBCs when tested in growth inhibition activity (GIA) assays *in vitro* [7,13,14]. However, when using these antigens as immunogens to induce similar antibody responses *in vivo*, the results have been less promising. One example of this is the blood stage-vaccine candidate PfRH5, a

merozoite antigen that has proven to have limited efficacy in clinical trials [15] in spite of being the target of highly inhibitory monoclonal antibodies [13,16].

The reasons why it is challenging to induce effective antibodies towards these antigens *in vivo* are not fully understood, but the magnitude and kinetic parameters of the induced antibody responses are critical factors. The PCRCR complex proteins have very short exposure times during infection, so both high antibody titers and fast on rates are crucial to block invasion by interfering with the binding of PfRh5 to Basigin on the RBCs surface [7].

Another important element contributing to the functional activity of merozoite-specific antibodies is complement activation. Several studies have shown that merozoite antigens activate complement [17–19] and, more recently, that complement fixing antibodies to these antigens correlate more strongly with protective immunity than growth–inhibitory antibodies [17,20].

However, *Plasmodium* parasites have evolved mechanisms to evade complement-mediated clearance, including the capture of human complement regulators. A few studies have shown that *P. falciparum* merozoites bind Factor H (FH) from human plasma [21,22]; this protein represents a major down-regulator of the alternative pathway (AP) of the complement. Moreover, Kennedy et al. demonstrated that merozoites exposed to human serum actively recruit C1 esterase inhibitor (C1-INH) to their surface, a down-regulator of the classical (CP) and lectin pathway (LP) of the complement [23]. In the same study, the authors identified the *P. falciparum* merozoite surface protein 3 (PfMSP3) as the binding partner for C1-INH. However, the real impact of this recruitment on complement activation in natural immunity towards malaria remains unclear. The rates of antibody acquisition to merozoite antigens vary in an antigen-specific manner over several exposures to the parasite [24,25]. Thus, it is likely that complement activation state or evasion could differ in the presence of merozoite-specific antibodies acquired early in life as compared to antibodies acquired later in life, after many encounters with Plasmodium parasites.

In this study, we use a cohort of plasma samples obtained from malaria convalescent children and from adult individuals with prior exposure to P. *falciparum* to examine complement activation and evasion by merozoites in the presence of naturally acquired antibodies.

We present novel insights on how activation of the complement system varies with the immune status of the host during the development of natural immunity to malaria. Moreover, we describe a monoclonal antibody that showed the ability to neutralize complement evasion by merozoites by interfering with the recruitment of the complement down-regulatory molecule C1-INH.

## Materials and methods

### Ethics statement

The institutional review boards of Noguchi Memorial Institute for Medical Research at the University of Ghana (NMIMR-IRB: CPN: 010/12–13) and the Ghana Health Service (GHS-ERC 08/05/14) approved the research ethics regarding human subjects. The use of plasma samples from non-exposed Danish naïve adults was approved by the Regional Research Ethics Committees for the Capital Region of Denmark (Protocol H-4-2013-083).

### Cohort samples

Plasma samples from children aged 1–12 diagnosed with clinical malaria were collected from May to June 2015 as part of a longitudinal cohort study at Hohoe Municipal Hospital, Ghana [26]. Samples were collected two weeks after the day of malaria diagnosis from all children and *P. falciparum* malaria was confirmed by a positive rapid diagnostic test (RDT) and by light microscopy examination of parasitemia (inclusion criteria: parasite density above 2500 parasites/µl of blood). The 39 children plasma samples used in this study were selected among all the convalescent children with a confirmed *P. falciparum* infection exclusively based on volume availability (plasma volume available >300 µl) and not based on disease severity or any other clinical or demographic characteristics. Of these 39 donors, 20 were diagnosed with severe *P. falciparum* malaria and 19 with uncomplicated *P. falciparum* malaria. An additional plasma sample was obtained four weeks post diagnosis for 16 of these children.

Plasma samples from 24 adults who were likely to have been exposed to *P. falciparum* multiple times were collected in Accra, Ghana, at the end of the rainy season (September) 2019. Ghana is a country with high transmission intensity during the rainy season spanning the months of April to August [27]. To exclude ongoing blood stage infection in the adult donors, all plasma samples were tested by both RDT and light microscopy and confirmed to be negative. The immune samples used in this study are representative of the geographical area in which they were collected.

Plasma from 8 malaria naïve adult individuals from Denmark were included as negative controls. Samples were collected after a written informed consent had been obtained from the subject or, in the case of children, from a parent or legal guardian. Demographic of all subjects and clinical characteristics of the convalescent children are listed in Table 1.

## Expression and purification of recombinant proteins

Recombinant PfRH5 was kindly provided by Prof. Simon J Draper (University of Oxford, UK) and was expressed in a Drosophila melanogaster cell line as described previously [28].

Plasmids encoding the recombinant merozoite antigens PfMSRP5-bio-his (Addgene plasmid #50805), PfSERA9-bio-his (Addgene plasmid #50820), PfMSP3-bio (Addgene plasmid #47731), PfRAMA-bio-his (Addgene plasmid #50737), Pf92-bio (Addgene plasmid #47728) and PfCyRPA-bio-his (PFD1130w-bio-his, Addgene plasmid #50823) were gifts from Dr. Gavin Wright, University of Oxford, Oxford, UK [29,30]. The plasmids were used for protein expression using the Exp293 expression system (Invitrogen), following manufacturer instructions. Briefly, 293F cells were cultured to the density of $3 \times 10^6$ cells per ml and transfected with the plasmids (1µg plasmid/ml of cell culture) using ExpiFectamine 293 Transfection kit (Invitrogen). Cell supernatants were harvested by centrifugation on day five post transfection.

The plasmid encoding recombinant RH4-his was a gift from Prof. Wai-Hong Tham (University of Melbourne, Australia). The protein was expressed in *E. coli* and purified as previously described with a second purification step using a SEC column (Superdex 200 GL, Cytiva) [31,32].

His-tagged proteins were purified using the ÄKTAxpress purification system (Merck), where the supernatant was passed through a 5mL HisTrap HP chromatography column (GE Healthcare #17524802). The proteins containing a ratCD4d3+4-tag only were purified using a prepacked HiTrap N-Hydroxysuccinimide -Activated HP 5 ml column, (GE Healthcare #17071701) with immobilized anti-rat CD4d3+4 antibody (OX68 hybridoma cell line, ECACC #94011007–1VL).

## Mice immunizations

Mice experiments were granted ethical approval by the Danish Animal Experiment Inspectorate (license number 2018-15-0201-01541, project approval number P24-13). Experiments were performed following the guidelines of the Federation of European Laboratory Animal Science Associations (FELASA).

**Table 1. Demographic and clinical characteristics of subjects.**

|  | Children | Adults | Controls |
|---|---|---|---|
| Age |  |  |  |
| Median (range) | 44 (1–12) | 37 (26–57) | 32(25–45)) |
| **Gender** |  |  |  |
| Male n. (%) | 18 (46%) | 19 (79%) | 3 (38%) |
| Female n. (%) | 21 (54%) | 5 (21%) | 5 (62%) |
| **Clinical category** |  |  |  |
| Severe *P. falciparum* malaria n. (%) | 20 (51%) | n. a. | n. a. |
| Uncomplicated *P. falciparum* malaria n. (%) | 19 (49%) | n. a. | n. a. |
| n.a. = non applicable |  |  |  |

BALB/c female mice (Janvier lab) were immunized with recombinant PfMSP3 protein (10µg/mouse) in 50% Addavax (InvivoGen). The vaccinations consisted of an initial prime, followed by three boosts with two weeks intervals. All injections were performed intramuscularly except for the last intraperitoneal boost three days before terminating the experiment and harvesting blood and spleens from terminally anesthetized mice.

## ELISA

To test the IgG reactivity of plasma samples, ELISA was performed as described previously [25]. Briefly, 96 well plates (Nunc MaxiSorp, ThermoFisher Scientific) were coated overnight with 2 µg/ml of recombinant protein. Plates were washed with PBS 0.05% tween20 (PBSt) and blocked for one hour (1h) using 5% skimmed milk diluted in PBSt. After three washes, plasma samples diluted 1:200 in blocking solution were added to the wells and incubated for 1hour. Plates were washed again and incubated for 1 hour with anti-human IgG-AP secondary antibody (Sigma Aldrich) diluted at 1:1500 in blocking buffer. Plates were developed in 4-Nitrophenyl phosphate disodium salt hexahydrate tablets (Sigma Aldrich) dissolved in Diethanolamine Substrate Buffer (Sigma Aldrich) and absorbance at 405 nm was measured using a micro-plate photo reader (Hipo MPP-96, Biosan). Antibody levels were presented as arbitrary units (AU) calculated as ($OD_{sample}$ - $OD_{blank}$)/ ($OD_{cutoff\ value}$ - $OD_{blank}$). The cut-off value for each antigen was calculated as the mean + 3*Standard Dev of the OD values of plasma from eight malaria-naïve Danish donors.

For the quantitative measurement of human C1-INH and FH from the immune plasma samples, we used the Human C1 inhibitor ELISA Kit (ab224883, abcam) and the Human Complement factor H ELISA Kit (ab252359, abcam) following the manufacturer´s instructions.

## Production of monoclonal antibodies

We have previously described the production of murine monoclonal antibodies to Plasmodium antigens [7,33]. In brief, splenocytes from mice immunized with recombinant PfMSP3 protein were fused with myeloma cells (SP2/0-Ag14) using ClonaCell-HY hybridoma cloning kit (Stemcell Technologies) according to the manufacturer's guidelines. After initial fusion and selection on semisolid HAT media, the clones were transferred to 96-well plates containing HT supplemented DMEM media containing 10% heat inactivated fetal calf serum (FCS) (Sigma-Adrich), 4 mM glutamine and penicillin/streptomycin. After three days the supernatants were screened by ELISA for specificity against PfMSP3. Positive clones were expanded, followed by single cell sorting into 96 well plates using a FACS Melody cell sorter (BD). The monoclonal cultures were expanded and gradually adapted to DMEM media with no HT supplement, supplemented with 10% HyClone low IgG FCS, glutamine, and penicillin/streptomycin. Antibodies from the cell supernatants were purified on a protein G column using the ÄKTAxpress purification system (Merck). We obtained a single monoclonal antibody against PfMSP3, named MP3.01.

To obtain IgG pools from immunized mice or plasma donors, equal volumes of serum from each mouse or plasma from each donor were mixed and diluted with PBS to 50 mL final volume prior to be loaded onto a 5 mL HiTrap Protein G column (Cytivalifesciences). The column was washed with PBS and IgG was eluted using 0.1 M glycine pH 2.7 and immediately neutralized with 1 M of TRIS pH 9.

## CryoEM analysis

MP3.01 IgG was digested with papain and purified by size exclusion chromatography on a Superdex 200 Increase 10/300 column (Cytiva). MP3.01 Fab and PfMSP3 recombinant protein were incubated together in a 1:1.1 ratio Fab:antigen with a total protein concentration of 1.2 mg/mL and, mixed with n-Octyl-beta-D-glucoside (OBG) for a final detergent concentration of 0.5 critical micelle concentration (CMC). From the mix, 3 µL were added to freshly glow discharged 1.2/1.3 300-mesh UltraAuFoil grids (EMS), the grids vitrified in liquid ethane using a Vitrobot Mark IV (ThermoFisher) operated at 4°C 100% humidity, and stored in liquid nitrogen until further use. The grids were imaged on a Glacios 2 transmission electron

microscope (Thermofisher) equipped with a Thermo Scientific Falcon 4i Direct Electron Detector. EPU was used to collect 12,458 movies at a total dose of 60 e-/Å2, a defocus range of -0.8 to -2.0 μm, a magnification of 195k and a voltage of 200 kV.

Movies were motion corrected and the Contrast Transfer Function (CTF) parameters calculated using Cryosparc 4.4.1 Live. Particles were selected using blob picker and extracted at a box size of 384, 320, and 256 px (S4 Fig). Pixel size was 0.718 Å. Particles were cleaned up using 2D classification and four distinct superclasses reextracted at 320 px and used to train four independent Topaz particle picking algorithms that were used to add additional particles [34]. Particles were used to generate five ab-initio models and the particles of the highest quality model were subjected to another round of 2D classification. The particles were used to generate ten ab-initio models and the well-defined model was used for non-uniform refinement with C1 symmetry. The AlphaFold2 [35] model of PfMSP3 was obtained from uniprot.org (entry A0A159SMZ5), the low confidence part trimmed away, and the high confidence part of the model fitted into the generated electron density map using ChimeraX [36].

### Parasite culture and merozoite purification

The *P. falciparum* laboratory-adapted strain 3D7 was maintained in culture using fresh O+ erythrocytes at 2% hematocrit. The culture medium was supplemented with 0.5% AlbuMAX (Life Technologies) as a substitute for human serum. One day prior purification of trophozoites, ring stage parasites were synchronized by treatment with 5% sorbitol. The Vario-MACS magnetic separation unit together with the MACS CS-column (Milteny Biotec) were used to separate the infected RBCs, containing trophozoites, from non-infected RBCs as previously described [37]. The purified trophozoites were cultivated in fresh complete medium until the parasites became highly segmented schizonts. Merozoites were subsequently obtained by filtering the schizonts through a 1.2 μm filter (Sartorius). The isolated merozoites were resuspended in 5% BSA in PBS and immediately used for either flow cytometry experiments or for parasite protein extraction.

### Surface staining and flow cytometry

Merozoites obtained from each T75 parasite flask set at 5% parasitemia were resuspended in 5 mL of 5% BSA/PBS (wash buffer), and 100 μL of this merozoite suspension was plated per well in a 96-well U-bottom plate. Merozoites were incubated at room temperature (RT) in 20% normal human serum (NHS) (Sigma Aldrich; S1), 20% FH-depleted serum (Complement Technology; A337), 20% human donor plasma or human purified complement protein at the desired concentration diluted in wash buffer. The purified complement proteins used in the experiments are FH (Complement Technology; A137) and C1-INH (Complement Technology; A140).

After incubation, merozoites were washed three times in wash buffer. They were subsequently stained with primary antibodies in wash buffer and incubated on ice for 20 minutes. Next, the merozoites were washed three times in wash buffer and, if the primary antibody was unconjugated, staining was performed with the relevant fluorophore-conjugated secondary antibody diluted in wash buffer. The wells were washed three times and fixed with 1% paraformaldehyde (PFA) supplemented with the live/death staining Hoechst 33342 (Invitrogen; H3570) (1:2000). Finally, the merozoites were resuspended in 200 μL wash buffer and run on the CytoFLEX (Beckman Coulter). FACS data were analyzed using FlowJo software (TreeStar).

All commercial primary antibodies were used in 1:200 dilution unless stated otherwise. Lab-produced monoclonal antibodies were used at the stated concentrations. Unconjugated antibodies used were: α-PfMSP3 mouse monoclonal antibody (clone MP3.01 produced in our lab), Factor H mouse monoclonal (Santa Cruz Biotechnology; C18/3; sc47685), C1 inhibitor rabbit polyclonal (ThermoFisher; PA5–31541) and Complement C9 Monoclonal Antibody (aE11) (Invitrogen; MA5–33373).

The fluorophore-conjugated secondary antibodies used to detect primary antibodies were Goat anti-Mouse polyclonal IgG PE (Abcam; Ab97041), Horse anti-Mouse IgG (H + L) FITC (Vector Laboratories; FI-2000), F(ab')2-Donkey anti-Rabbit

polyclonal IgG (H + L) PE (Invitrogen; 12-4739-81). Directly conjugated antibodies used in this study were goat polyclonal α-Human IgG Fc PE (Invitrogen; 12-4998-82), PE α-human IgM Antibody (Biolegend; 14507) and C1q Rabbit polyclonal FITC (Abcam; Ab4223). A monoclonal antibody against the merozoite surface antigen PfMSRP5 (produced in our lab) was used as positive control to stain compensation samples.

## Protein extraction and Western Blot analysis

Isolated merozoites were incubated for 20 minutes in either wash buffer, wash buffer with 20% NHS. The parasites were subsequently spun down at 3000xg for 5 minutes at 4°C and the pellet was washed in PBS to remove unbound serum proteins. Hereafter, the merozoites were lysed by adding RIPA buffer (Thermo Scientific) supplemented with protease inhibitor cocktail (Roche Diagnostics). The parasites were disrupted by vortexing every 2 minutes for 15 minutes while incubated on ice. Finally, the lysates were centrifuged for 15min. at 15000xg at 4°C and the supernatant was stored at -80°C until use. For SDS-PAGE, equal volumes of lysate were loaded on NUPAGE 4–12% gradient Bis-Tris gel (Invitrogen) with MOPS as running buffer. Samples were prepared in loading buffer under non-reducing conditions and heated to 95°C for 10min before loading. Using a wet-blotting system (Novex X-Cell II, Invitrogen), the proteins were transferred to a 0.45 μm nitrocellulose membrane. After blocking with 5% skimmed milk in PBS, human FH and Human C1-INH were detected using anti-FH HRP-conjugated antibody (Santa Cruz Biotechnology; sc-166613 HRP) and anti-C1-INH rabbit polyclonal (ThermoFisher; PA5–31541). Two control parasite proteins were also detected: the heat-shock protein 70 (PfHSP70) using anti-PfHSP70 rabbit polyclonal antibody (SPC-186; StressMarq Biosciences; 1:3000) and cysteine-rich protective antigen (PfCyRPA), using anti-PfCyRPA mouse monoclonal antibody clone Cyp1.9 produced in our lab (https://doi.org/10.3389/fimmu.2021.716305). Goat anti-Rabbit HRP or Rabbit anti-Mouse HRP (DAKO; 1:2000) were used to detect the unconjugated primary antibodies. After washing the membrane, blots were developed by LumiGLO peroxidase chemiluminescent substrate (SeraCare) and images were acquired using the ChemiDoc MP Imaging System (Bio-rad).

## Growth inhibition activity assay

Parasite cultures of *P. falciparum* 3D7 strain were synchronized by treatment with 5% sorbitol one day prior to assay setup. GIA assays were performed in sterile 96 well half-area microplates (Corning; 3696), as previously described [7,38,39]. Prior to their use in the assay, the plasma samples were depleted of erythrocyte-specific antibodies by incubation for 1 hour at room temperature with 50μl fresh 0 + erythrocytes/mL plasma. On the day of the assay, synchronized mid-trophozoites were adjusted to 0.5% parasitemia and incubated for approximately 42 hours (1 parasite life cycle) with the relevant plasma samples at 1:15 dilution in parasite media supplemented with NHS. Each plasma sample or control mAb dilution was tested in triplicates. The one-cycle GIA assay was harvested when parasites reached early schizont stage. A biochemical measurement using the *P. falciparum* lactate dehydrogenase assay was used to quantify parasitemia. The results of the GIA assays, expressed as growth inhibition percentage, were calculated as follows:

$$GIA\% = 100 - \left[ \frac{(A630\ \textit{Inhibited sample} - A630\ \textit{RBCs only})}{(A630\ \textit{Uninhibited sample} - A630\ \textit{RBCs only})} \right] \times 100$$

For each plate we included a serial dilution of a previously described neutralizing antibody specific for PfRH5 and calculated the EC50 value of its sigmoidal inhibition curve to control for assay-to-assay variation [7,13]

## Statistical Analysis

Data analysis was performed and graphs created using GraphPad Prism Software (version 10, GraphPad). Mann-Whitney test was used to determine differences in merozoite antigen-specific IgG levels between children and adults plasma groups measured by ELISA. Differences in GIA between children plasma samples from week 2 and week 4 post

diagnosis were also analyzed by Mann-Whitney test. Kruskal-Wallis and Dunn's post-hoc test was used to evaluate differences between children, adults and dk controls plasma groups in GIA% or expression of surface-bound molecules in FACS experiments. Linear regression analysis was used to measure the relationship between C1q binding and IgG or IgM binding to merozoites as well as the correlations between antigen (Pf92 and PfMSP3)-specific IgG levels and complement down-regulators (FH and C1-INH) recruitment.

## Results

### Plasma from convalescent children induce superior parasite growth inhibition compared to plasma from exposed adults

In areas with endemic P. *falciparum* infection, resistance to severe malaria is generally acquired by the age of five years and clinical immunity to blood stage infection is achieved by adulthood [3,40]. Given the central role of antibodies in protection from blood stage parasites, studying the different features of the antibody response in exposed children and adults is critical for a better understanding of malaria immunity. Thus, using samples from convalescent children (day 14 after a malaria diagnosis), Ghanain adults and Danish malaria-naïve controls, we tested the ability of plasma to inhibit the growth of P. *falciparum* 3D7 parasites using the *in vitro* GIA assay. Plasma samples were depleted of RBC-specific antibodies prior to use in the assay. Strikingly, parasite growth inhibition above the 20% negative cut-off value was observed for 32 out of the 39 (82%) children's samples, while only 1 out of the 24 (4, 2%) adult samples was slightly inhibitory (Fig 1A). The mean percent inhibition for the children's samples was 59.2% (± 28.4%) while around zero for the adult samples (Fig 1B). As expected, none of the samples from malaria naïve individuals inhibited the growth of the parasite (Fig 1A and 1B). Interestingly, the growth inhibition capacity of the children plasma samples did not directly reflect disease severity, as the range of the GIA % values was similar between the severe malaria and the uncomplicated malaria group (S1 Fig).

Given the striking difference between the children and the adults samples in inhibiting parasite growth, we next used ELISA to test IgG specific responses towards a panel of P. *falciparum* merozoite antigens (PfMSP3, Pf92, PfRAMA, PfCyRPA, PfRH4, PfRH5, PfMSRP5 and PfSERA9) (Fig 1C). We confirmed the presence of IgG specific for these antigens in both groups of samples and observed different binding profiles, depending on the specific antigen rather than on the plasma group. For example, we found that PfRAMA and PfCyRPA-specific IgG responses were of a greater magnitude in children compared to adults, while PfMSP3-specific IgG reactivity was significantly higher in adults (Fig 1C). However, when looking at the antibody prevalence in the two groups of donors, IgG prevalence was higher in children compared to adults for six out of the eight antigens tested (Pf92, PfRAMA, PfMSRP5, PfSERA9, PfCyRPA, PfRH4; Fig 1C).

To functionally characterize the IgG present in the children immune plasma, we aimed at repeating the GIA assay using pure IgG fractions. Due to the limited volume of the plasma samples, we could not attempt IgG purification from each individual. Instead, we extracted the total IgG fraction from a pool consisting of plasma from the six donors that showed the highest GIA above 90% (high inhibitory). The pooled purified IgG obtained from the high inhibitory children was tested in the GIA assay in a dilution series, starting at the concentration of 10 mg/mL. A previously tested inhibitory IgG fraction that was obtained from a pool of immune individuals from Liberia [41] served as positive control. The IgG from the high inhibitory children pool showed measurable GIA at the two highest concentrations tested (10 and 5 mg/ml) similarly to the positive control IgG, albeit with slightly lower GIA% values (Fig 1D). These data confirm that, as expected, immune IgG contributes to the GIA observed within the children plasma samples.

For 16 of the convalescent children, we had access to an additional plasma sample collected 4 weeks after diagnosis of malaria. We performed GIA assays using these samples and observed a significant reduction in parasite growth inhibition when samples from the same donors were taken 4 weeks after the diagnosis compared to 2 weeks (Fig 1E).

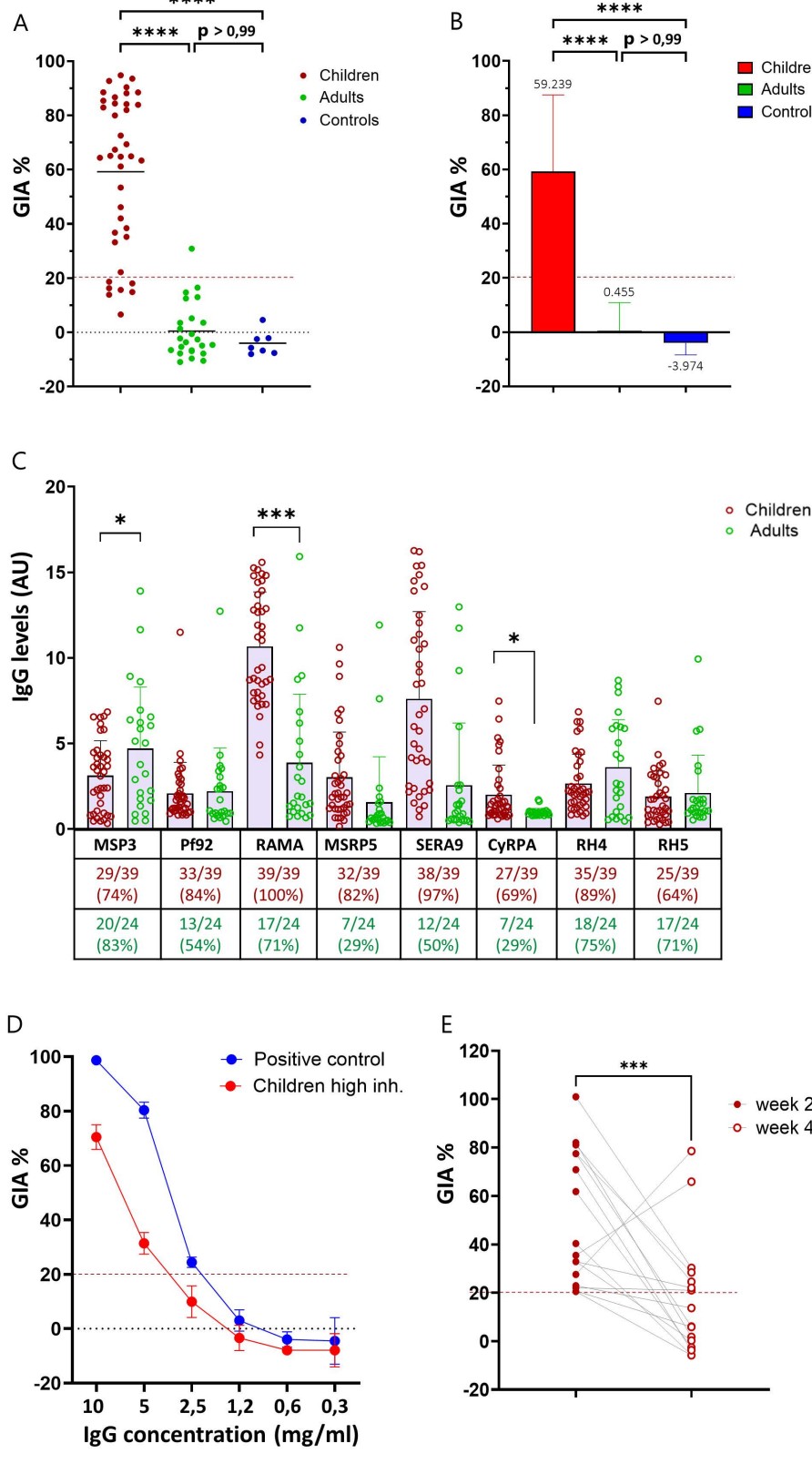

**Fig 1. Parasite growth inhibition capacity and IgG reactivity of plasma from convalescent children and exposed adults. A)** GIA assay of immune plasma samples (children and adults) and naïve plasma samples (controls) tested at 1:30 dilution on *P. falciparum* 3D7 parasites. Data depicted

as percentage of growth inhibition (GIA%) calculated as described in the Methods section. The dotted line at 20% growth inhibition indicates the assay cut-off below which samples are considered non-inhibitory. **B)** Mean % inhibition of the three plasma samples groups from A. **C)** ELISA detecting plasma IgG reactivity towards the *P. falciparum* merozoite recombinant antigens MSP3, Pf92, RAMA, MSRP5, SERA9, CyRPA, RH4, RH5. IgG levels are expressed as arbitrary units (AU) obtained by dividing the OD values of the immune plasma samples by the cut-off value of each given antigen. The cut-off value was calculated for each antigen as the mean + 3SD of OD values of plasma from eight naïve plasma donors. The numbers in the table indicate the prevalence of the antigen-specific IgG within the donor group (number of donors with a positive IgG response/total number of donors and %). **D)** GIA assay on 3D7 *P. falciparum* parasites using dilution series of purified IgG fraction obtained from a pool of the 6 top high inhibitory children plasma samples. The IgG positive control sample represents IgG fractions from immune individulas from Liberia previously shown to have high GIA on 3D7 parasites when tested at the concentration of 10 and 5 mg/ml. **E)** GIA assay of immune plasma samples collected from 16 of the convalescent children 4 weeks after malaria diagnosis and of matching donors smples from week 2 after diagnosis. Samples are tested at the dilution of 1:30 on *P. falciparum* 3D7 parasites. The dotted line at 20% growth inhibition indicates the assay cut-off. The asterisks indicate statistical significance calculated using Kruskal-Wallis test in panels A, B and E and Mann-Whitney test in panel C (*p < 0.05, **p < 0.01, ***p < 0.001, ****p < 0.0001).

### Plasma from children and adults induce similar initiation of the classical complement pathway but different MAC deposition on merozoites

Previous work from our lab has shown that, in convalescent children, merozoite-specific IgG levels stay relatively high four weeks after infection while IgM responses to the same antigens have already declined [25]. Therefore, we analyzed the binding of IgG, IgM and complement activation proteins to the parasite in the presence of immune plasma. We purified *P. falciparum* merozoites and incubated them with the individual plasma samples from the three cohorts, as well as with NHS, prior to surface staining for IgG, IgM and C1q bound to the parasite surface by Flow Cytometry. The mean fluorescence intensity (MFI) for each staining is expressed as fold over NHS control (MFI after incubation of merozoites in NHS). This enabled us to compare results from separate experiments. We found that while IgG binding to merozoites was similar in the presence of plasma from children or adults, IgM binding to the parasite was significantly higher in the presence of children plasma compared to plasma from adults (Fig 2A and 2B). However, when looking at C1q deposition on the surface of live merozoites incubated in immune plasma, we observed similar MFI values for the children and adult group (Fig 2C). As expected, binding of IgG, IgM and C1q to the merozoite surface was significantly lower in the presence of naïve plasma as compared to immune plasma, with MFI values similar to those observed in the presence of NHS (fold over NHS control≈1, Fig 2A, 2B and 2C). These results suggest that antibodies from convalescent children and naturally immune adults fixate C1q and activate the upstream part of the classical complement pathway in a similar way. Notably, we observed some degree of binding of IgG, IgM and C1q to the parasite also in the presence of non-immune plasma (both naïve controls and NHS) (Fig 2D, 2E and 2F). However, this was significantly lower than that observed in the presence of plasma from malaria exposed individuals (Fig 2A, 2B and 2C). Linear regression analysis of C1q binding versus IgG or IgM binding to merozoites showed a significant positive correlation between C1q and IgG binding for the adults, while for the children that was not significant (S2A Fig). No correlation was observed between IgM and C1q binding to merozoites (S2B Fig).

Next, we aimed to examine the activation of the downstream part of the complement pathway induced by naturally acquired antibodies. Antibody-mediated complement activation leads to deposition of the membrane attack complex on the surface of pathogens, causing disruption of the cell membrane, cell lysis and death [42]. Therefore, we examined the presence of the MAC on the surface of merozoites after incubation with immune plasma as well as with naïve control plasma. To this end, we incubated purified live merozoites with the individual plasma samples prior to surface staining for the MAC (c5b-9) deposited on the cell surface. Strikingly, we found a significantly higher deposition of the MAC in the presence of children plasma as compared to plasma from adults (Fig 2G and 2H). Thus, antibodies present in plasma from convalescent children are more effective in activating the downstream part of the complement cascade than antibodies from exposed adults. However, for both children and adult plasma samples, the fluorescent intensity of the MAC staining was higher than that of naïve controls, albeit the difference between exposed adults and controls did not reach statistical significance. (Fig 2G).

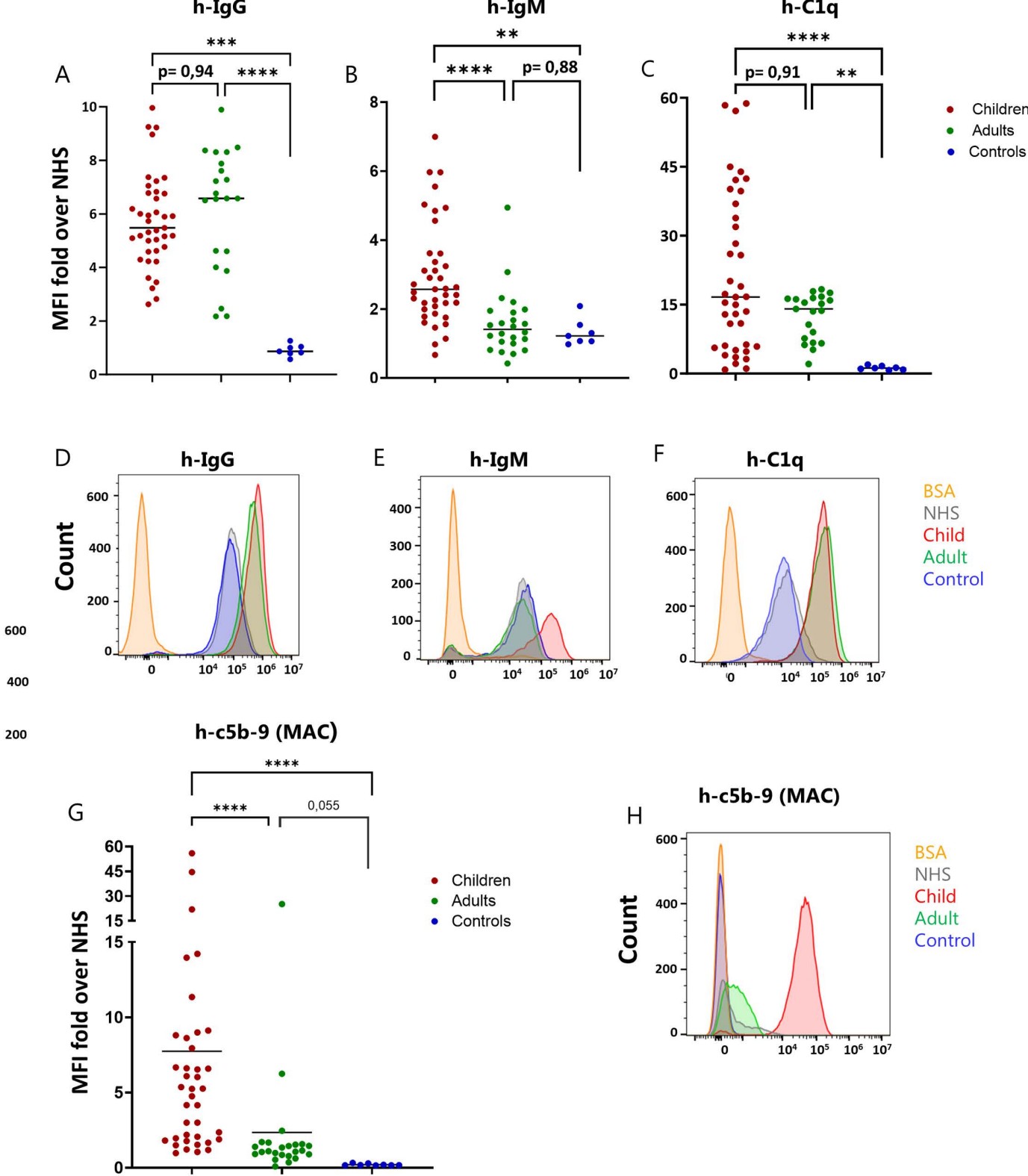

**Fig 2. Upstream and downstream activation of the complement classical pathway induced by immune plasma from children and adults.** Flow cytometry detection of human IgG (A), human IgM (B) and human C1q (C) bound to the surface of 3D7 merozoites after incubation with immune plasma

from children or adults as well as naïve plasma (controls). The data are expressed as MFI fold over NHS (MFI of the staining with each plasma donor/ MFI of the staining with NHS). Each donor sample is depicted as an individual dot. Representative histogram plots for the staining of human IgG (D), IgM (E) and C1q (F) on the surface of merozoites after incubation with plasma from one of the immune (child and adult) or naïve (control) donors as well as NHS. G) Flow cytometry detection of human c5b-9 (MAC) bound to the surface of 3D7 merozoites after incubation with immune plasma from children or adults as well as naïve plasma (controls). The data are expressed as MFI fold over NHS (MFI of the staining with each plasma donor/MFI of the staining with NHS). Scattered plots depict each donor as an individual dot. H) Representative histogram plots for the staining for human c5b-9 on the surface of merozoites after incubation with plasma from one of the immune (child and adult) or naïve (control) donors as well as NHS. The asterisks indicate statistical significance calculated using Kruskal-Wallis test (**$p < 0.01$, ***$p < 0.001$, ****$p < 0.0001$).

### Antibodies from convalescent children can block FH recruitment by merozoites, restoring activation of the complement alternative pathway

Given that plasma from children and adults could similarly induce C1q deposition (suggesting activation of the classical pathway) but lead to a different downstream complement activation state, we next looked at the effects of the malaria-immune plasma on regulators of the alternative pathway.

Several studies have shown that *P. falciparum* merozoites can actively recruit FH from human plasma to their surface to evade recognition by the complement alternative pathway [21,22]. However, the effects of this recruitment on complement activation state downstream as well as the significance of this immune evasion mechanism in naturally acquired immunity to malaria remain elusive.

First, we wanted to confirm that we could detect FH bound to the surface of merozoites after incubation with human plasma or serum. Live merozoites were incubated in NHS or in PBS-BSA prior to surface staining for FH and analysis by Flow Cytometry. In parallel, protein extracts were prepared from merozoites incubated with NHS or PBS-BSA and FH was detected by Western Blot analysis. As expected, we observed FH deposition on the cell surface when parasites had been in contact with NHS but not with PBS-BSA (Figs 3A and S3A). Then, we wanted to investigate how recruitment of human FH was affected by the presence of naturally acquired antibodies from convalescent children and immune adults. Therefore, live merozoites were incubated with individual plasma samples from the three groups in our study and FH bound to the cell surface was detected by flow cytometry. The MFI of FH staining after incubation with NHS was again used to normalize the data. We found that, in the presence of plasma from convalescent children, the MFI of FH staining was significantly reduced compared to that observed in the presence of naïve plasma (Fig 3B). Interestingly, FH staining after incubation with the adult plasma samples was not significantly different from that of naïve controls (Fig 3B). These results suggest that merozoite-specific antibodies present in convalescent children are more effective than antibodies present in immune adults in blocking FH binding to the parasite. To ensure that our results were not influenced by differences in plasma levels of FH between children and adults, we measured the concentration of FH in all the plasma samples by quantitative ELISA and confirmed that the two groups of samples contained similar amounts of circulating FH (S3B Fig).

The merozoite antigen Pf92, a member of the six-cysteine (6-cys) protein family, has been suggested as the binding partner of human FH on the merozoite surface [21]. Thus, we hypothesized that antibodies directed against Pf92 in the immune plasma could compete with FH binding on the parasite surface. We performed linear regression analysis of the FH recruitment measured by flow cytometry (FH MFI) combined with the levels of Pf92-specific antibodies detected by ELISA (α-Pf92 IgG AU) (Fig 3C). For both the children and the adult group there was a tendency towards a negative correlation between Pf92 antibody levels and FH recruitment, however it was not statistically significant (Fig 3C). To confirm the link between FH binding to the parasite and the inhibition of the MAC deposition, we purified live merozoites and incubated them with NHS or with human serum that was depleted of FH (FHd-HS) but contained all the other complement molecules. As expected, no FH was recruited to the parasite surface after incubation with FHd-HS (Fig 3D). MAC deposition was indeed increased when FH was absent from human serum, as demonstrated by the higher MFI of the staining for c5b-9 on the merozoite surface after incubation with FHd-HS as compared to NHS (Fig 3E). Addition of soluble factor

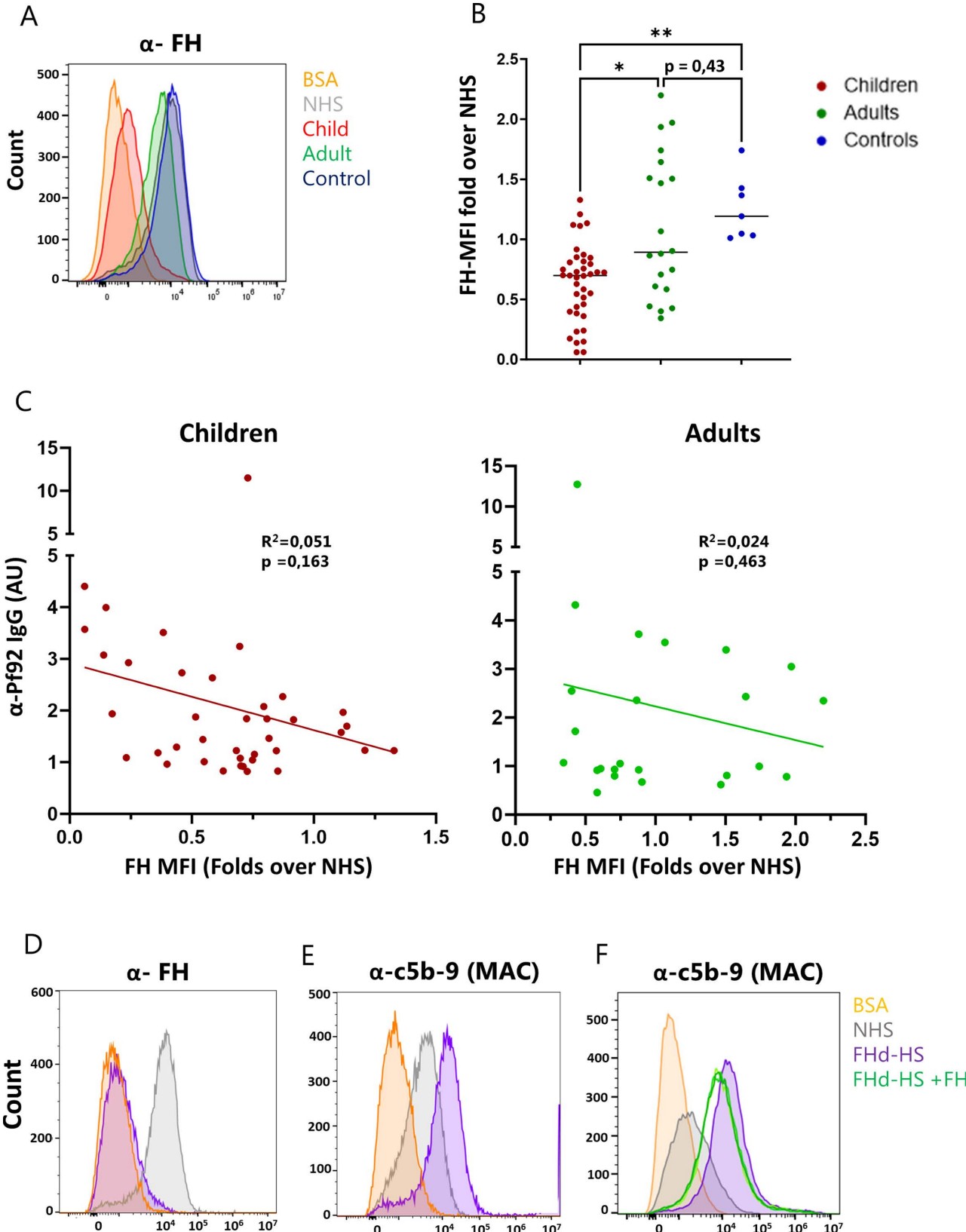

**Fig 3. FH recruitment and MAC deposition to the merozoite surface induced by immune plasma. A)** Flow cytometry detection of human FH bound to the surface of 3D7 merozoites after incubation with immune plasma from children or adults as well as naïve plasma (controls). The data are

expressed as MFI fold over NHS (MFI of the staining with each plasma donor/MFI of the staining with NHS). Scattered plots depict each donor as an individual dot. **B)** Representative histogram plots for the staining of human FH on the surface of merozoites after incubation with plasma from one of the immune (child and adult) or naïve (control) donors as well as NHS. **C)** Correlation between plasma IgG reactivity towards Pf92 (α-Pf92 IgG OD) and FH deposition on merozoites (FH MFI) using plasma samples from the children (red dots) and adults (green dots). Correlation coefficient ($R^2$) and p value are shown in the plot. **D)** Representative histogram plots for the staining of human FH on the surface of merozoites after incubation with normal serum (NHS) or serum depleted of FH (FHd-HS). **E)** Representative histogram plots for the staining of human c5b-9 (MAC) deposited on the surface of merozoites after incubation with normal serum (NHS) or serum depleted of FH (FHd-HS). **F)** Representative histogram plots for the staining of human c5b-9 (MAC) deposited on the surface of merozoites after incubation with normal serum (NHS), serum depleted of FH (FHd-HS) or serum depleted of FH supplemented with soluble FH (FHd-HS + FH) at the concentrations of: 10 μg/ml (lightest green), 50 μg/ml (light green) and 100 μg/ml (dark green). The asterisks indicate statistical significance calculated using Kruskal-Wallis test (*$p < 0.05$, **$p < 0.01$).

FH to FHd-HS reduced the MFI of the staining for c5b-9, albeit not to the levels observed with NHS (Fig 3F). However, we must stress that in this experiment we could not use an amount of FH high enough to reach the average plasma levels of 250 μg/ml, due to limits in the concentration of the purified protein (Fig 3F).

**Naturally acquired antibodies are ineffective in blocking the binding f C1-INH to merozoites**

We had observed that *P. falciparum* merozoite-specific antibodies developed in children could re-activate the complement cascade by presumably competing with FH binding to the parasite surface, thus counteracting immune evasion. We then aimed at performing a similar analysis on immune evasion of the classical and the lectin pathways in the presence of naturally acquired antibodies.

The fluid-phase molecule C1-INH controls the classical and lectin pathways by inactivating the proteases C1r, C1s, MASP1, and MASP2 [43]. This complement inhibitor was shown to bind *P. falciparum* merozoites via the parasite surface antigen PfMSP3 [23]. First, we confirmed the presence of C1-INH on the surface of merozoites following incubation with NHS, regardless of the complement activation state, consistent with previous report [23]. This was done using both flow cytometry analysis of C1-INH bound to the surface of live merozoites, and Western Blot analysis of protein extracts from merozoites that were pre-incubated with NHS. (Figs 4A and S3A). We then addressed if naturally acquired antibodies from convalescent children and immune adults influenced the parasites ability to recruit human C1-INH. Live merozoites were incubated with individual plasma samples from the three groups in our cohort and C1-INH bound to the cell surface was detected by flow cytometry. The MFI of C1-INH staining after incubation with NHS was again used to normalize the data. Interestingly we found that, in contrast to our observations with FH, the presence of plasma from convalescent children enhanced the binding of C1-INH to the merozoites, as the MFI of the staining for this molecule was significantly higher when the parasite was incubated with the children plasma samples compared with the adult plasma samples (Fig 4A and 4B). Nevertheless, C1-INH recruitment to the cell surface was slightly higher in the presence of the adult plasma compared to the naïve plasma for most of the adults samples, although the difference did not reach statistical significance (p value = 0.063) (Fig 4B). These results suggest that antibodies acquired during the development of natural immunity are not capable of competing with the binding of C1-INH to PfMSP3. However, when quantifying this complement inhibitor in all the plasma samples by quantitative ELISA, we observed significantly higher concentrations in children plasma compared to adult plasma (Fig 4C). This was not entirely surprising as C1-INH is an acute-phase protein, with serum levels rising about 2-fold during infection and inflammation [44]. Thus, plasma samples from convalescent children still contained elevated levels of C1-INH due to the recent Plasmodium infection, which may account for the increased binding of C1-INH to merozoites observed in the samples from children.

To circumvent this problem and ensure equal starting concentration of C1-INH in our assay, we used purified immune and naïve IgG fractions to pre-incubate live merozoites prior to exposure to human purified C1-INH at a fixed concentration. Surface staining followed by Flow Cytometry analysis confirmed increased binding of C1-INH to merozoites in the presence of malaria-immune IgG compared to naïve IgG, the latter showing a fluorescence intensity of the C1-INH

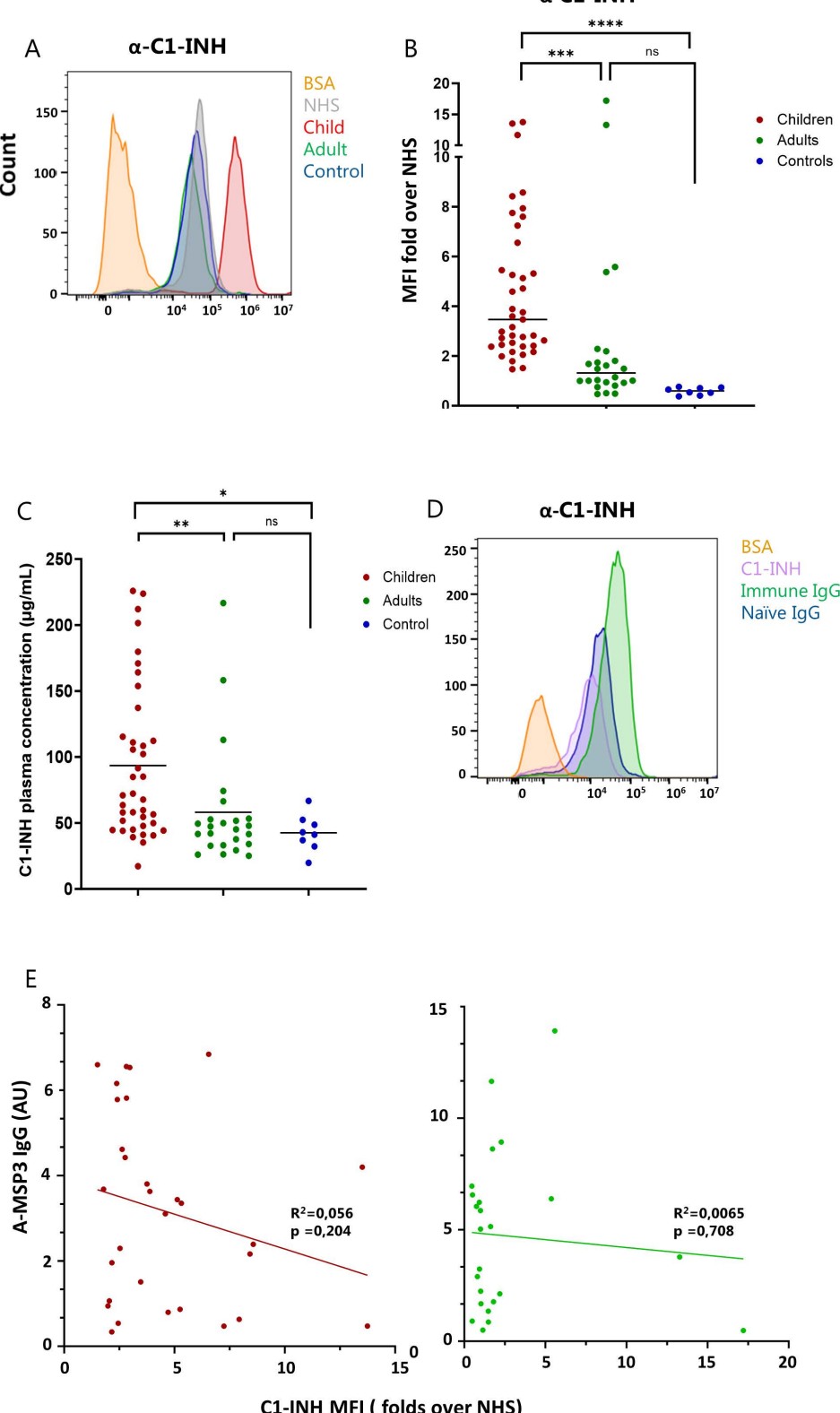

**Fig 4. Recruitment of C1-INH by merozoites in the presence of naturally acquired antibodies of children and adults. A)** Flow cytometry detection of human C1-INH bound to the surface of 3D7 merozoites after incubation with immune plasma from children or adults as well as naïve plasma

(controls). The data are expressed as MFI fold over NHS (MFI of the staining with each plasma donor/MFI of the staining with NHS). Scattered plots depict each donor as an individual dot. **B)** Representative histogram plots for the staining of human C1-INH on the surface of merozoites after incubation with plasma from one of the immune (child and adult) or naïve (control) donors as well as NHS. **C)** Quantitative ELISA measuring the plasma levels of C1-INH in the immune plasma samples as well as the naïve Danish controls. C1-INH concentration in each plasma sample is expressed as µg/mL, calculated by interpolation from a standard curve. **D)** Immune and naive IgG fractions purified from donor plasma pools are used to pre-incubate live merozoites prior to exposure to human purified C1-INH (IgG from immune adults in green, IgG from naïve dk controls in blue). C1-INH binding to the parasite surface is detected by Flow Cytometry. Representative histogram plots for the staining for C1-INH with and without IgG pre-incubation are depicted. The asterisks indicate statistical significance calculated using Kruskal-Wallis test (*p < 0.05, **p < 0.01, ***p < 0.001, ****p < 0.0001).

staining similar to that observed in the absence of IgG preincubation (Fig 4D). These data confirm that naturally acquired antibodies towards malaria merozoites cannot compete with the binding of C1-INH to the parasite, but rather enhanced the recruitment. It is noteworthy that antibodies specific for PfMSP3, the binding partner of C1-INH on the merozoite surface, were detected in plasma samples from both children and adults (Fig 1C). However, these antibodies seem to not be effective in interfering with the binding of complement regulators. Linear regression analysis of the PfMSP3-specific antibody response (α-pfMSP3 IgG) versus the recruitment of C1-INH to the parasite (C1-INH MFI) showed no significant correlation between the two parameters (Fig 4E).

The mechanisms behind the enhanced C1-INH binding to merozoites in the presence of immune plasma or purified IgG remain to be addressed.

### Targeting PfMSP3 with a mouse mAb drastically reduced the binding of C1-INH to the merozoite surface

To investigate further whether targeting PfMSP3 with antibodies other than those developed after natural infection could potentially decrease C1-INH recruitment to the parasite, we generated a mouse monoclonal antibody (mAb MP3.01) against this merozoite surface antigen using hybridoma technology. Polyclonal antibodies were further purified from the serum obtained from PfMSP3 immunized mice. The antibodies were tested for their ability to interfere with the recruitment of human purified C1-INH to the surface of live merozoites by Flow Cytometry. We found that both the mAb and the polyclonal IgG targeting PfMSP3 were capable of decreasing the binding of C1-INH to the merozoites (Fig 5A and 5B). However, the MP3.01 mAb was more effective than the polyclonal IgG, at least when tested at equal concentrations (50 µg/ml), as the fluorescent intensity of the C1-INH staining after pre-incubation with the mAb was reduced to background level (BSA) (Fig 5A). Moreover, the mAb´s ability to block C1-INH recruitment to the parasite surface was dose-dependent, with maximum effect observed when used at the concentration of 50µg/ml (Figs S4A and 4C). The specificity of the MP3.01 mAb blocking was confirmed using an isotype control mouse mAb as negative control. (S4E Fig). The polyclonal α-PfMSP3 IgG, on the other hand, showed only partial inhibition of C1-INH recruitment, even when tested at the concentration of 500µg/ml (Figs S4B and 3D). Thus, we generated a mAbs that could compete with the binding of a complement down-regulator to its merozoite target antigen more effectively than naturally acquired antibodies.

However, the MP3.01 mAb did not have any effect on the binding of C1-INH to the merozoite surface in the presence of NHS (S5A Fig) or immune plasma (S5B and S5C Fig). This may be due to changes in the way C1-INH binds to PfMSP3 in the presence of immune IgG/IgM and C1q, something that requires further investigation.

In an effort to characterize the epitope on PfMSP3 binding to the MP3.01mAb - which could possibly be the same involved in the binding to C1-INH - the recombinant antigen and the Fab region of MP3.01 were complexed and examined through cryoEM. We obtained a low-resolution model where PfMSP3 was found to fold into a three-alpha helix bundle with the MP3.01mAb binding site at the end (Figs 5C-D and S6). The model reached 3.7 Å at a Gold Standard Fourier Shell Correlation cutoff of 0.143 and had a resolution range of 3–6 Å. This is, to our knowledge, the first report of structural data for PfMSP3. The electron density map is available on the Electron Microscopy Data Bank (EMDB) under accession code EMD-51634.

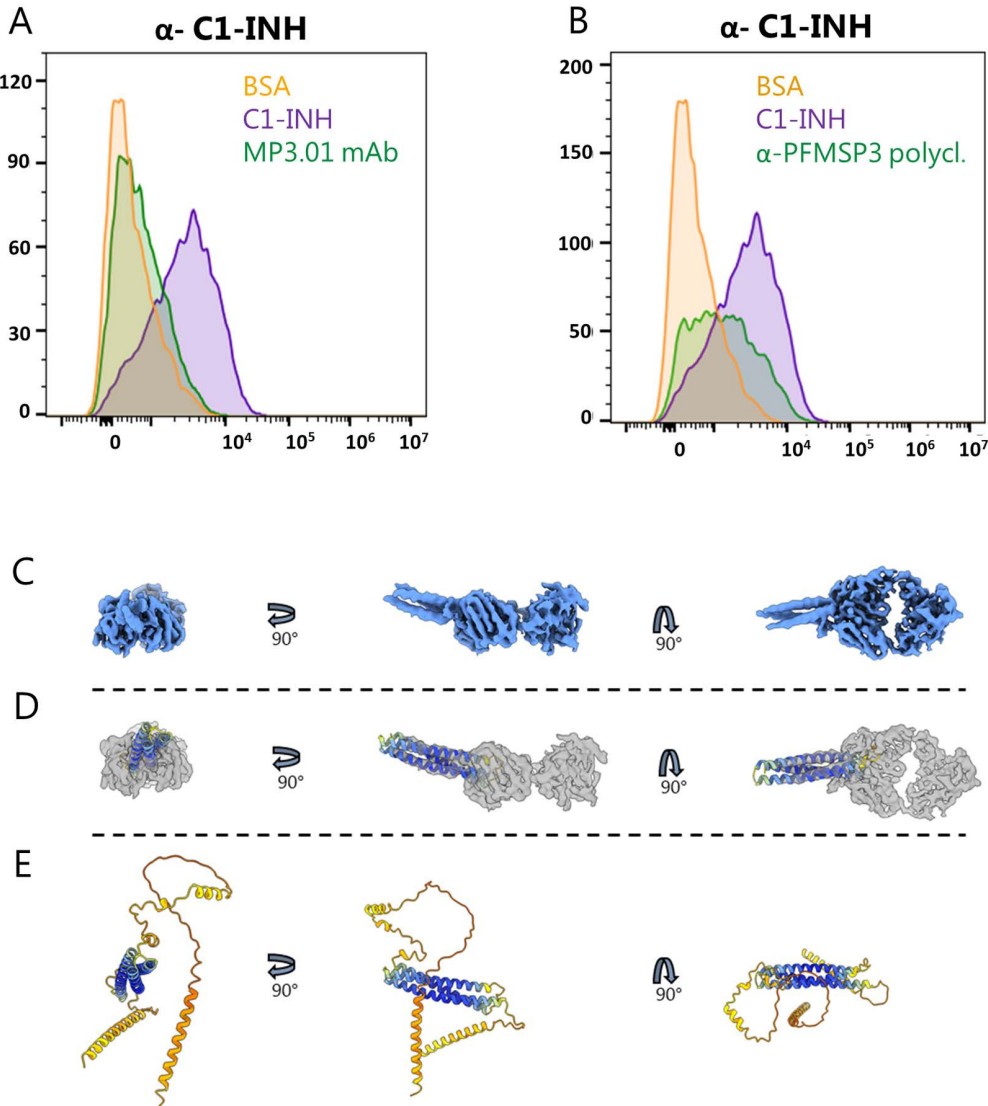

**Fig 5. The mouse mAb MP3.01 targeting PfMSP3 prevents the binding of C1-INH to the merozoite surface. A)** Flow cytometry analysis of human C1-INH binding to the parasite surface with or without pre-incubation of merozoites with the mouse mAb MP3.01 (orange: merozoites incubate in wash buffer; purple: merozoite incubated with 0,02 µg/µl of C1-INH; green: merozoite pre-incubated with 50 µg/mL of MP3.01 mAb prior to addition of 0,02 µg/µl of C1-INH). Representative histogram plots for the staining of C1-INH are depicted. **B)** Flow cytometry analysis of human C1-INH binding to the parasite surface with or without pre-incubation of merozoites with the mouse polyclonal IgG raised against PfMSP3. (orange: merozoites incubate in wash buffer; purple: merozoite incubated with 0,02 µg/µl of C1-INH; green: merozoite pre-incubated with 50 µg/mL of MP3.01 polyclonal IgG prior to addition of 0,02 µg/µl of C1-INH). Representative histogram plots for the staining of C1-INH are depicted. **C)** Electron density map of PfMP3 in complex with the Fab region of MP3.01. **D)** The high confidence part of the AlphaFold2 model of PfMSP3 superimposed on the three putative alpha helices of the electron density map. **E)** The full AlphaFold2 model of PfMSP3.

## Discussion

In this study, we present new insights on how naturally acquired antibodies towards merozoites modulate complement system activation in different stages of the development of immunity towards malaria. We observed that plasma from convalescent children had superior parasite inhibition capacity than plasma from Ghanaian adults, which does not reflect the clinical manifestation of malaria in the two groups of individuals. As the adults are protected from severe disease, our

findings suggest that the *in vivo* protective efficacy of the antibodies present in these individuals relies on effector functions, like Fc-mediated functions engaging complement and innate cells, rather than on blocking invasion. Our data are in line with a previous and, to our knowledge, only study on this topic showing that plasma-mediated growth inhibition of blood stage *P. falciparum* decreases with age [45]. These findings are relevant when considering the use of the GIA assay data as immunological surrogates of protection in natural infection and suggest that additional assays are needed to predict the *in vivo* efficacy of antibodies on blood stage parasites, especially when comparing different age groups.

Here, we found that parasite inhibition by the children plasma is significantly decreased four weeks after diagnosis, a time point when merozoites-specific IgM responses are no more detectable in the children plasma while IgG levels to the same antigens are still stable [25]. The role of IgM in protecting convalescent children from severe disease is still debated, but previous studies showed that IgM fractions from plasma of malaria-exposed children inhibited parasite growth *in vitro* [46,47], supporting our speculation that IgM could contribute to the high parasite inhibition of children plasma. However, here we did not succeed in purifying the IgM fraction for testing it in our experiments, thus we can´t rule out the contribution of other acute phase plasma molecules to the higher GIA of children plasma collected 2 weeks post diagnosis.

We confirmed that live merozoites recruit complement down-regulatory molecules to their surface to evade immune recognition, as previously reported [21–23]. However, these previous studies did not investigate complement features in the presence of naturally acquired antibodies, but only in the presence of NHS and a single monoclonal antibody. Thus, our work adds valuable knowledge on host complement activation and evasion by the malaria parasite during natural infection.

When looking at the upstream activation of the classical pathway of the complement, we found similar IgG binding and C1q deposition on merozoites in the presence of natural antibodies from children and adults. On the other hand, antibodies from children induced enhanced MAC deposition on the parasite surface compared to antibodies from adults, and thus had a higher level of downstream complement activation. Moreover, antibodies from convalescent children were more effective in reducing the binding of the down-regulator FH to the merozoites as compared to antibodies developed in adults with prior exposure to the parasite. The features determining the capacity of children´s antibodies of disrupting the binding between FH and Pf92 on the merozoite surface remain to be elucidated. However, our data suggest that it is not just higher antibody titers induced by the recent boost of infection in the convalescent children, as the magnitude of the plasma IgG reactivity towards Pf92 was found to be similar in the two groups (Fig 1C).

In contrast, when looking at the classical and lectin pathway regulator C1-INH, natural antibodies could not interfere with its binding. This may be due to several factors, such as insufficient titers, binding to a different region of PfMSP3 than the one involved in the binding to C1-INH and the high abundance of C1-INH in human plasma (0.15–0.30 mg/mL).

One finding that surprised us was the enhanced C1-INH binding to merozoites in the presence of immune plasma. The mechanisms behind this remain to be addressed, though we speculate that this may be partially due to the presence of C1q on the merozoite surface in the presence of immune plasma, as this molecule is an additional target for C1-INH binding [48,49]. However, this does not explain why C1-INH binding to merozoites was enhanced also in the presence of purified immune IgG fraction that should not contain C1q.

In the attempt of counteracting immune evasion of the classical complement pathway by the parasite, we generated a murine mAb targeting PfMSP3- the merozoite binding partner for C1-INH. Strikingly, this mAb was able to abolish the recruitment of C1-INH to the merozoite surface, unlike antibodies found in naturally exposed individuals.

These findings serve as proof of concept that vaccination could potentially induce PfMSP3-specific antibodies more effective than naturally acquired ones in neutralizing complement evasion via competing with C1-INH recruitment to the parasite surface.

Using cryo-EM, we obtained a low-resolution electron density map of PfMSP3 in complex with the Fab of MP3.01. PfMSP3 took the shape of a three-alpha helix bundle with a highly similar torsion to the bundle predicted by high certainty by AlphaFold2. The rest of the protein was in the examined context not resolved, indicating disordered regions that may

only be organized upon interaction with other proteins of the MSP3 protein complex, such as MSP1, MSP6 and MSP7 [50]. MP3.01 was found to bind at the end of the three-alpha helix bundle of PfMSP3, indicating a potential binding site for C1-INH in the same region, as the mAb competed with the complement regulator in binding to the merozoite surface. These findings can guide epitope-based vaccine design of PfMSP3 as part of a multi-component malaria vaccine.

Our data suggest that targeting parasite antigens binding to complement down-regulators with vaccination could represent a valid approach to enhance the efficacy of future malaria vaccines. We therefore suggest including these antigens, together with leading blood stage antigens, in novel multicomponent vaccines to obtain superior antibody effector functions through the re-activation of the complement system. Re-activation of complement *in vivo* during blood stage malaria could be beneficial to many aspects of host immunity. Beyond pathogen lysis via MAC deposition and inhibition of RBCs invasion, active complement would promote phagocytosis by immune cells and immune activation. Targeting complement evasion has been successfully exploited to design a vaccine against *Neisseria meningitides,* where the bacteria binding partner of human FH is encoded as vaccine antigen [51,52]. We anticipate that this strategy could be further utilized in the design of vaccines towards other infectious diseases where complement recognition is evaded with similar mechanisms.

### Study limitations

We acknowledge that the plasma samples from convalescent children and exposed adults used in this study were collected from two different geographical areas at different time points, which could influence some of the results presented here. However, we must stress that the aim of this study is to compare antibodies from recently exposed individuals (convalescent children) to antibodies from individuals who had been exposed in the past (exposed adults). We present evidence that the superior ability to compete with the binding of FH by the antibodies from children compared to the antibodies from adults is not just a reflection of higher antibody response and/or seroprevalence, as the binding of total plasma IgG to live merozoites as well C1q deposition on the parasite surface were similar in the two groups. The main difference we observe between the two groups is the higher IgM binding to merozoites in the presence of plasma from the children, which reflects their recent exposure to the parasites.

A limitation of this study is that the number of individuals in our cohort is relatively small, thus follow-up studies on a larger sample group are needed to corroborate and expand our findings. Furthermore, the murine monoclonal antibody MP3.01 showed no reduction of C1-INH recruitment to the merozoite surface in the presence of NHS/plasma, thus we could not address whether this mAb would enhance MAC deposition on the parasite surface. This is something we aim to investigate further, as we believe it may be due to changes in the way C1-INH binds to its specific binding partner on the surface of pathogens in the presence of immune IgG/IgM and C1q, something that has still not fully understood.

### Supporting information

**S1 Fig. Parasite growth inhibition capacity of plasma from children with severe malaria (SM) and uncomplicated malaria (UM).** Data from Fig 1A are plotted dividing the children donors in the two clinical categories.
(TIF)

**S2 Fig. Relationship between deposition of C1q and binding of merozoite-specific IgG and IgM to merozoites.** Correlation between IgG **(A)** or IgM **(B)** binding and C1q binding to merozoites after incubation with immune plasma from children (top panels, red dots) and adults (bottom panels, green dots) is measured by simple linear regression analysis. Correlation coefficients ($R^2$) and p values are depicted in the plots.
(TIF)

**S3 Fig. Western blot detection of FH and C1-INH binding to merozoites and plasma FH levels measured by ELISA. A)** Western blot analysis to detect human FH, human C1-INH and the parasite control proteins PfHPS70 and PfCyRPA in protein extracts prepared from merozoites incubated with BSA control buffer (-NHS) or with NHS (+NHS).

Arrows indicate protein bands corresponding to the expected size. **B)** Quantitative ELISA measuring the plasma levels of FH in all the immune plasma samples. FH concentration in each plasma sample is expressed as µg/mL, calculated by interpolation from a standard curve.
(TIF)

**S4 Fig. C1-INH binding to merozoites after pre-incubation with PfMSP3-specific murine antibodies at different concentrations. A, B)** Flow cytometry analysis of human C1-INH binding to the parasite surface with and without pre-incubation of merozoites with different concentrations of MP3.01 (A) or PfMSP3 specific polyclonal IgG (B). Representative histogram plots for the staining of C1-INH are depicted. Orange curve: merozoites incubated with BSA; purple curve: merozoites incubated with C1-INH; green curves: merozoites pre-incubated with α-PfMSP3 antibodies prior to addition of C1-INH (antibody concentrations specified next to the curves). **C, D)** MFI values of the surface staining for human C1-INH on the merozoites surface after pre-incubation with different concentration of MP3.01 (from A) and PfMSP3 polyclonal IgG (from B) The dashed orange line corresponds to the background MFI for C1-INH obtained after staining of merozoites incubated in BSA wash buffer. **E)** Flow cytometry analysis of human C1-INH binding to the parasite surface after pre-incubation of merozoites with MP3.01 (50 µg/mL, green curve) or 9AD4 isotype control mAb (50 µg/mL, pink curve). Merozoites incubated with human purified C1-INH (purple curve) or BSA wash buffer (orange curve) serve as positive and negative controls, respectively.
(TIF)

**S5 Fig. C1-INH binding to merozoites in the presence of NHS or immune plasma with and withouth pre-incubation with MP3.01-mAb. A)** Flow cytometry analysis of human C1-INH binding to the parasite surface with (dotted gray line) or without (gray line) pre-incubation with 50 µg/mL of MP3.01 mAb prior to addition of 20% NHS. Representative histogram plots for the staining of C1-INH are depicted. **B)** Flow cytometry analysis of human C1-INH binding to the parasite surface with (dotted red line) or without (red line) pre-incubation with 50 µg/mL of MP3.01 mAb prior to addition of 20% pooled plasma from children. Representative histogram plots for the staining of C1-INH are depicted. **C)** Flow cytometry analysis of human C1-INH binding to the parasite surface with (dotted green line) or without (green line) pre-incubation with 50 µg/mL of MP3.01 mAb prior to addition of 20% pooled plasma from adults. Representative histogram plots for the staining of C1-INH are depicted.
(TIF)

**S6 Fig. Workflow for data processing of PfMSP3 in complex with Fab MP3.01 and achieved resolution. A)** Workflow of generation of PfMSP3:MP3.01 electron density map. Bar on representative micrograph denotes 80 nm. Box size of 2D classes are equal to 28, 23 and 18 nm for box sizes of 384, 320, and 256 px, respectively. **B)** Gold Standard Fourier (GSC) shell correlation and local resolution of the electron density map.
(TIF)

**S7 Fig. Gating strategy.** Gating for merozoites on forward versus side scatter followed by gating for Hoechst positive live cells before analysis of C5b-9 fluorescence. These plots are representative for the gating strategy of all FACS experiments.
(TIF)

## Acknowledgments

The authors wish to thank Prof. Simon J Draper (University of Oxford, UK) for providing us with recombinant PfRH5 and Prof. Wai-Hong Tham (University of Melbourne, AUS) for the gift of the plasmid encoding recombinant PfRH4. We would like to express our gratitude to Prof. Michael Theisen (University of Copenhagen and Statens Serum Institut, DK) for the IgG sample from naturally immune individuals from Liberia. Moreover, we thank Lisa Eshun-Wilson (The

Scripps Research Institute, USA) for assistance in the data processing of the cryoEM data of PfMSP3 and MP3.0. Lastly, the authors would like to thank Maiken Visti (University of Copenhagen) for her extensive work and parasite expertise.

## Author contributions

**Conceptualization:** Maria Rosaria Bassi, Bogdan Cristinoi, Melanie Rose Walker, Lea Barfod.

**Data curation:** Maria Rosaria Bassi, Bogdan Cristinoi, Frank Buitenwerf, Mark Bergholt Cuadrado, Kasper Haldrup Björnsson, Andrew B. Ward.

**Formal analysis:** Maria Rosaria Bassi, Frank Buitenwerf, Mark Bergholt Cuadrado, Kasper Haldrup Björnsson.

**Funding acquisition:** Kasper Haldrup Björnsson, Lea Barfod.

**Investigation:** Bogdan Cristinoi, Frank Buitenwerf, Mark Bergholt Cuadrado, Lea Barfod.

**Methodology:** Maria Rosaria Bassi, Bogdan Cristinoi, Frank Buitenwerf, Mark Bergholt Cuadrado, Kasper Haldrup Björnsson, Melanie Rose Walker, Frederica Dedo Partey, Andrew B. Ward, Michael Fokuo Ofori, Lea Barfod.

**Project administration:** Lea Barfod.

**Resources:** Frederica Dedo Partey, Michael Fokuo Ofori.

**Software:** Maria Rosaria Bassi, Bogdan Cristinoi, Frank Buitenwerf, Mark Bergholt Cuadrado, Kasper Haldrup Björnsson, Andrew B. Ward.

**Supervision:** Maria Rosaria Bassi, Melanie Rose Walker, Andrew B. Ward, Lea Barfod.

**Validation:** Maria Rosaria Bassi, Frederica Dedo Partey, Michael Fokuo Ofori.

**Writing – original draft:** Maria Rosaria Bassi, Bogdan Cristinoi.

**Writing – review & editing:** Maria Rosaria Bassi, Bogdan Cristinoi, Frank Buitenwerf, Kasper Haldrup Björnsson, Melanie Rose Walker, Frederica Dedo Partey, Lea Barfod.

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
