## [Decision Letter · Decision Letter 0]

8 Nov 2024

PPATHOGENS-D-24-02068Deposition of complement regulators on the surface of Plasmodium falciparum merozoites depends on the immune status of the hostPLOS Pathogens Dear Dr. Barfod, Thank you for submitting your manuscript to PLOS Pathogens. After careful consideration, we feel that it has merit but does not fully meet PLOS Pathogens's publication criteria as it currently stands. Therefore, we invite you to submit a revised version of the manuscript that addresses the points raised during the review process. Please submit your revised manuscript within 60 days Jan 07 2025 11:59PM. If you will need more time than this to complete your revisions, please reply to this message or contact the journal office at plospathogens@plos.org. Please include the following items when submitting your revised manuscript:* A rebuttal letter that responds to each point raised by the editor and reviewer(s). You should upload this letter as a separate file labeled 'Response to Reviewers '. This file does not need to include responses to any formatting updates and technical items listed in the 'Journal Requirements' section below.* A marked-up copy of your manuscript that highlights changes made to the original version. You should upload this as a separate file labeled 'Revised Manuscript with Track Changes '.* An unmarked version of your revised paper without tracked changes. You should upload this as a separate file labeled 'Manuscript '. If you would like to make changes to your financial disclosure, competing interests statement, or data availability statement, please make these updates within the submission form at the time of resubmission. Guidelines for resubmitting your figure files are available below the reviewer comments at the end of this letter. We look forward to receiving your revised manuscript. Kind regards, James G. Beeson, MBBS, PhDAcademic EditorPLOS Pathogens Dominique Soldati-FavreSection EditorPLOS Pathogens Michael Malim

Editor-in-Chief

PLOS Pathogens

orcid.org/0000-0002-7699-2064   **Journal Requirements:** **Additional Editor Comments (if provided):** The reviewers have raised a number of important issues for you to consider. Please carefully address their concerns as you revise the manuscript.**Reviewers' Comments:** Reviewer's Responses to Questions

**Part I - Summary**

Reviewer #1: Dear Editor,

Thank you very much for the opportunity to review the paper “Deposition of complement regulators on the surface of Plasmodium falciparum merozoites depends on the immune status of the host” by Bassi, M.R and colleagues.

In this paper the authors present a very interesting body of work investigating the role of host immune factors, specifically antibody acquisition with age, on recruitment and deposition of complement regulatory factors recently shown to be involved in parasite immune evasion to the surface of P. falciparum merozoites. Increasing evidence in the field of malaria immunity has shown that beyond direct antibody neutralisation (mediating growth inhibition) that has been shown inconsistent associations/correlation with protection from malaria, other antibody Fc-mediated functions including complement fixation on merozoite surfaces play an important role in immunity against malaria and might be more strongly correlated with protection than growth inhibition and for example greater growth inhibition is seen in the presence of antibodies plus complement compared to antibodies alone. However, recent studies have shown that P. falciparum appears to have evolved mechanisms to avoid these effector function by recruiting complement regulatory factors (Factor H (FH) and C1 esterase inhibitor (C1-INH)) to the surface of merozoites that inhibit/dampen complement activation but, unfortunately, these studies were done in the absence of naturally acquired antibodies, a major gap in our understanding of the interaction between parasites, host immunity and complement system. The authors here address this important question by investigating recruitment of these complement regulatory proteins in the presence of immune antibodies from cohort studies in Ghana and report several major findings that included important differences in immune acquisition and outcomes between children and adults including:

1. Antibodies from children induced greater growth inhibition that that of adults.

2. Antibodies from children were more effective at mediating the terminal complement cascade MAC formation compared to antibodies from adults despite similar levels of complement C1q, the initial component of the antibody dependent complement cascade

3. Factor H recruitment to the surface of merozoites was significantly lower in the presence of antibodies from children compared to adults that showed no difference with malaria naïve samples. These results suggest blocking of FH recruitment on merozoite surface by naturally acquired antibodies.

4. But for C1-1NH this was not the case with antibody samples from children shown to enhance binding of C1-1NH on merozoites suggesting they don’t compete for C1-1NH binding as for FH.

5. A monoclonal antibody against PfMSP3 inhibited/decreased binding on C1-1NH to the surface of merozoites, a fact that was not seen with naturally acquired antibodies with the authors suggesting that vaccines could be developed that are more efficient and inducing antibodies more efficient at inhibiting complement evasion by the parasites than that seen with naturally occurring antibodies.

Reviewer #2: This study emphasizes that antibodies from convalescent children recovering from malaria are notably effective in disrupting the parasite's ability to evade the immune system. These antibodies reduce the recruitment of complement regulatory proteins, particularly Factor H, which controls the alternative complement pathway. Interestingly, the study finds that while children's antibodies are more effective at hindering Factor H recruitment, antibodies from both children and adults enhance C1-INH binding (controlling the classical and lectin pathways of complement activation), The authors developed a monoclonal antibody (MP3.01) targeting PfMSP3, a parasite antigen that binds to C1-INH and aids immune evasion. This antibody blocks C1-INH binding, unlike the naturally acquired antibodies measured, which could enhance vaccine efficacy by reducing C1-INH recruitment on the merozoite surface.

Overall the manuscript was well written and clear, and I appreciate the work going into developing a mAb against PfMSP3. The study lacks an experiment that directly demonstrates whether the use of this antibody leads to improved complement activation by decreasing C1-INH recruitment. Although the structural analysis of PfMSP3 bound to the MP3.01 antibody was provided, the paper did not include functional assays to examine the effect of reduced Factor H deposition (or C1-INH deposition) on complement activation merozoite lysis. The inclusion of such experiments would significantly strengthen the paper.

**Part II – Major Issues: Key Experiments Required for Acceptance**

Reviewer #1: Major comments

The major findings showing an increased inhibitory effect of FH recruitment and MAC formation by antibodies from children compared to adults while very interesting raise questions about the nature of the antibody samples in children versus adults used in this study which are my major comments/concerns in relation to the manuscript. From the cohort description provided here, lines 121 – 138, it appears very likely that these findings are an over interpretation of what could simply be explained by a huge bias of the choice of samples used and compared here and a detailed and major clarification and justification is needed to support these findings that otherwise cannot be justified. Specifically, the authors describe that plasma samples were collected from 39 children aged 1-12 years in 2015 from Hohoe in Ghana while 24 adult samples were collected from Accra Ghana in 2019.

1. From reference #26 given for the children cohort study, it appears this was a much larger study involving more than 100 children. What was the criteria used to then down-select the 39 children used here from the larger pool of 108 children that met the criteria of parasite density>2500 parasites/μl of blood? Are these 39 children part of the severe malaria (n=48) or uncomplicate malaria (n=60) group described in that study as this will greatly impact antibody responses seen? Are there any particularly important differences in the included compared to the excluded samples that could be biasing these results? A detailed explanation and comparison of different clinical and demographic details need to be given and presented of the included versus excluded samples to allay any fears of sample selection bias.

2. No details or references have been given for the adult study samples other than that one short sentence in lines 130-131 which is rather disappointing and very frustrating when describing a cohort study and made it hard to make any reasonable judgements on the choice of these samples as was possible from the looking up the reference given for the children study sample in (1) above. The authors need to clearly outline and clarify these details as stated in the comment above for children study samples.

3. It is rather concerning that major conclusions are being derived comparing two groups of samples, children versus adults, that were collected from two very geographically distinct locations; Hohoe, a mostly rural area in the North Volta region (children) versus Accra, the capital and an urban area in the Southern Coastal region (adults) of Ghana. Looking at the official “Malaria Indicator Surveys” from 2016 and 2019 from the National Malaria Control Program and other bodies in Ghana, malaria prevalence (as measured in children 6-59 months) was among the highest in the Volta region at 20% and lowest nationwide (2%) in Greater Accra region in 2019 and 28% versus 5% in 2016 suggesting significant differences in malaria exposure with greater exposure to the children from Hohoe that might explain their significantly higher seroprevalence of antibodies to the difference merozoite antigens tested and in turn greater capacity as well to mediate MAC formation and inhibition of FH recruitment. It is highly likely, therefore, that these conclusions are purely a result of differences in exposure and therefore antibody acquisition by region.

4. Additional to (3) above these two different groups of samples were collected over two very different time periods experiencing significant changes in malaria transmission and prevalence; children in 2015 and adults in 2019. Again the “Malaria Indicator Survey” state that “The percentage of children under age 5 testing positive for malaria according to microscopy has decreased consistently over time, from 27% in 2014 to 21% in 2016 and 14% in 2019”. It would, therefore, not at all be surprising to see lower immune responses in the adult group (from 2019) compared to the children group (from 2014) even if they were to have been collected from the same geographical region, which they were not.

Reviewer #2: - Sampling issues: A major issue with interpretation throughout the manuscript is that children and adult samples were collected from different locations and years, with a four-year gap. This could affect antibody quantities and quality, as the circulating parasites may differ over time, influencing antibody pools/subtypes/and test results. For example, It’s difficulty to interpret the difference in GIA observed between the post-malaria children and adults, given that the samples were taken from different areas of Ghana, where there is substantial district-level differences in malaria risk (see PMID 38594716). The adult samples were taken from Accra, where malaria transmission may be lower than in other rural areas. Some of these differences are revealed in the antibody binding measures shown in Fig 1C, where responses to a small panel of antibodies are measured. The discussion section needs a limitation section where this addressed, and claims regarding differences between “children” and immune adults need to be tempered throughout the manuscript. (e.g., Lines 508-510).

- Relatedly: How are the adults known to be “clinically immune?” There is no data about whether the adults have evidence of clinical immunity, only that they live in Accra. Would change the language from “clinically immune adults” to “Ghanian adults” or something similar throughout.

- There are a relatively small number of samples studied. 39 children (16 convalescent follow-up samples) and 24 adults. This needs to also be addressed in the discussion.

- Figure 1A: How was the 20% cut-off for the assay defined?

- Figure 1C: The Y-axis should be labeled as "IgG" instead of "Ig" to avoid confusion that it is total Ig classes. Could the authors also plot the data from the malaria-naïve controls? And was merozoite-specific IgM measured? Could that data be included?

- Figure 2: Why didn't/couldn’t the authors separate IgG from IgM to determine which is responsible for C1q and MAC deposition in children, especially given that IgM levels are high in convalescent children? This is mentioned in the discussion (line 682-685) but would be helpful to raise when these results are first described.

- Figure 2G: What do "dk controls" refer to? Would allso include statistical comparison between children vs. controls as in 2A.

- Figure 3: The source and specifications of the human serum used in that experiment should be provided. Figures 3D and E could benefit from using Factor H (FH)-depleted normal human serum (NHS), and a rescue experiment adding incremental amounts of FH to FH-depleted serum could determine its impact on MAC deposition.

- Correlation with IgM: In Figure 3C, it would have been helpful to examine the correlation between FH deposition and anti-Pf92 IgM levels, not just IgG.

- MAC Deposition in Figure 4: Did the authors assess MAC deposition in Figure 4 the same way it was done in Figure 3 (D-E), comparing levels in purified IgG with and without C1-INH? Could that be provided?

- MP3.01 Addition: In Figure 5, it would be interesting to see whether adding the MP3.01 antibody to serum from children, adults, or NHS could reduce C1-INH recruitment.

**Part III – Minor Issues: Editorial and Data Presentation Modifications**

Reviewer #1: Minor comments

1. How old were the adults included in the study? Give age range details as shown for children.

2. In the text, Figures 3B and 4B are mentioned before 3A and 4A; lines 497/506 and 551/561. Swap the Figures in the actual plots.

3. Lines 510-514 – so was the source of FH from the NHS or immune plasma samples? Or both? Did they measure FH in control samples too?

4. The suggestion that antibodies to Pf92 might compete with FH binding in parasite surface in lines 519-524 would be better shown with comparisons of levels of Factor H here with this Pf92 antibody between children and adults as in Figure 3A instead of correlation plots. It is not quite clear how different they are with the way they have presented it but looking at the data in the current format it looks like they might not be that different.

5. The authors show in lines 524-531 and Figure 3E that Factor H interferes with MAC formation with the assay done with NHS or factor H depleted NHS. Seeing this study is looking at the role of antibodies it would have been helpful/interesting to see these two comparisons made in the presence of children versus adult antibodies as well.

6. Figure 5A-B – what does the C1-NH group represent? It wasn’t clear from the text or figure legend.

7. The use of the term efficacy in regard to growth inhibition and other antibody assay endpoints is rather incorrect/misleading and should be restricted to protection/clinical outcomes for vaccine trials – e.g. lines 75, 88, 116, 612, 649, 666, 673, 701.

8. Lines 678-681 – interesting outcomes on IgM that also supported by another earlier group that deserves referencing (Boyle et al, PMID: 31579826).

9. Format reference 55.

10. Line 773 – “These findings”.

11. Could the authors speculate on the positive correlation seen between IgG and C1q in adult samples but not in children samples despite similar levels of C1q deposition in both groups.

12. Similar to testing of GIA with purified children (pooled) IgG as shown in Figure 1D it would have been helpful to have a comparison of the same with the adult samples, for example using a pool of the highest responders from the adults group.

13. It was not clear what results are being referred to by the huge amount of text from lines 415-427 as this reads more like a discussion of previously published work and would probably be best to be removed or significantly cut down to a sentence.

14. For all figures, check image qualities and make font sizes legible in places where they are tiny including numbers in graphs e.g. Figure 1

Reviewer #2: - Throughout – suggest changing language from “malaria infection” to “Plasmodium infection” or “malaria” since malaria is the disease and not what causes infection. (e.g., line 61, line 115, etc).

- Line 80: might soften this language, given a recent preprint that found that Rh5.1/Matrix M might indeed have significant protection in clinical trials (https://www.medrxiv.org/content/10.1101/2024.10.15.24315473v1)

- Line 95: Define FH (first time in main text, although it is also defined in abstract)

- In the results- lines 415-427 (*beginning with “previous work from our lab.”). This seems to be a justification for a set of experiments comparing binding, growth inhibition, and complement fixation between IgG and IgM. I was expecting data here but there are none (these data appear to be shared in the next results section.) Suggest removing these sentences here and starting the next section with a truncated version of this to help with narrative flow of the manuscript.

- Suggest removing ns from figures and putting in actual p-values

- First three paragraphs of the discussion seem redundant to the introduction? Could potentially just start with line 659?

PLOS authors have the option to publish the peer review history of their article (what does this mean? ). If published, this will include your full peer review and any attached files.

**Do you want your identity to be public for this peer review?** For information about this choice, including consent withdrawal, please see our Privacy Policy .

Reviewer #1: No

Reviewer #2: No

---

## [Decision Letter · Decision Letter 1]

19 Mar 2025

PPATHOGENS-D-24-02068R1

Deposition of complement regulators on the surface of Plasmodium falciparum merozoites depends on the immune status of the host

PLOS Pathogens

Dear Dr. Barfod,

Thank you for submitting your manuscript to PLOS Pathogens. After careful consideration, we feel that it has merit but does not fully meet PLOS Pathogens's publication criteria as it currently stands. Therefore, we invite you to submit a revised version of the manuscript that addresses the points raised during the review process.

Please submit your revised manuscript within 30 days May 18 2025 11:59PM. If you will need more time than this to complete your revisions, please reply to this message or contact the journal office at plospathogens@plos.org. Please include the following items when submitting your revised manuscript:

We look forward to receiving your revised manuscript.

Kind regards,

James G. Beeson, MBBS, PhD

Academic Editor

PLOS Pathogens

Dominique Soldati-Favre

Section Editor

PLOS Pathogens

Sumita Bhaduri-McIntosh

Editor-in-Chief

PLOS Pathogens

orcid.org/0000-0003-2946-9497

Michael Malim

Editor-in-Chief

PLOS Pathogens

orcid.org/0000-0002-7699-2064

**Additional Editor Comments :**

Please address the remaining minor requests from Reviewer 1 to make some further minor clarifications in the manuscript regarding differences between the children and adult cohorts.

Additionally, in the Abstract you draw comparisons between children and adult samples - you need to make a qualification here that children and adults were taken from different endemic regions since this points was highlighted as an important limitation by both reviewers.

In the abstract, the following sentence needs revision 'In particular, we show that, while the ability to fix C1q and activate the classical pathway is similar for antibodies deriving from convalescent children and clinically immune adults, downstream complement activation shown as deposition of the membrane attack complex (MAC) is strikingly higher with antibodies developed in convalescent children.' (clarify that the subjects come from 2 different regions; higher in convalescent children compared to who?)

**Journal Requirements:**

1) The following files are currently uploaded as file type 'Other', which are not viewable by the reviewers: (Data S1 GIA Fig 1A.xlsx,Data S2 ELISA Fig 1C.xlsx, Data S3 GIA Fig 1E.xlsx,Data S4 IgG GIA Fig 1D.xlsx, and Data S5 ELISA Fig 4C.xls). Please change the file types to 'Supporting Information' and include legends in the manuscript if you wish them to be included in review.

**Reviewers' Comments:**

Reviewer's Responses to Questions

**Part I - Summary**

Reviewer #1: Thank you for the opportunity to review the revised manuscript by Bassi, M.R. and colleagues titled "Deposition of complement regulators on the surface of Plasmodium falciparum merozoites depends on the immune status of the host"

The authors address important gaps in our understanding of how Plasmodium falciparum has evolved to evade important protective antibody Fc functions, specifically complement attack, by studying for the first time these interactions in the presence of antibodies against P. falciparum merozoites and revealing their impact on recruitment of complement proteins needed for immune evasion. They compared antibody responses from differently exposed individuals (children versus adults). Antibodies from children, but not adults, interfered Factor H recruitment (mediator of immune evasion) despite similar levels of complement C1q fixation and a monoclonal antibody to PfMSP3 blocked recruitment of C1-INH, another downregulator of the complement cascade mediating evasion, to the parasite surface suggesting a role for vaccination to induce antibodies to specific antigens that limit recruitment of certain complement proteins and in turn parasite immune evasion.

Reviewer #2: The authors have addressed the concerns raised in the prior round of review. I remain concerned about the cohort comparisons, but the authors have tempered their findings and adjusted language throughout the manuscript

**Part II – Major Issues: Key Experiments Required for Acceptance**

Reviewer #1: I appreciate efforts made by the authors to address the reviewer comments and at this stage only have minor comments.

1. While I appreciate the authors adding details clarifying on the selection criteria for the 39 donor samples, could they also explicitly add the point they mention in the responses to reviewers that these samples were (randomly) selected purely based on volume availability. This point has not been added to the current corrections.

2. Under the now added "study limitations" section, after the first sentence (lines 847-850), consider also adding briefly in the actual manuscript text the points they have mentioned in their responses to reviewers, point #3, stating that their comparison was between recently (children) and previously exposed individuals (adults) (also raised in response #4), and IgG to live merozoites and C1q deposition were similar in the two populations. They are important points that should be incorporated in the discussion text (not just responses to reviewers) so kindly confirm this has been captured well.

Reviewer #2: n/a

**Part III – Minor Issues: Editorial and Data Presentation Modifications**

Reviewer #1: Check throughout the manuscript for typos for example lines #19, 26, 53, 585

Reviewer #2: (No Response)

PLOS authors have the option to publish the peer review history of their article (what does this mean? ). If published, this will include your full peer review and any attached files.

**Do you want your identity to be public for this peer review?** For information about this choice, including consent withdrawal, please see our Privacy Policy .

Reviewer #1: No

Reviewer #2: No

**Figure resubmission:**
---

## [Editor Report · Decision Letter 2]

7 Apr 2025

Dear Dr. Barfod,

We are pleased to inform you that your manuscript 'Deposition of complement regulators on the surface of Plasmodium falciparum merozoites depends on the immune status of the host' has been provisionally accepted for publication in PLOS Pathogens.

Best regards,

James G. Beeson, MBBS, PhD

Academic Editor

PLOS Pathogens

Dominique Soldati-Favre

Section Editor

PLOS Pathogens

Sumita Bhaduri-McIntosh

Editor-in-Chief

PLOS Pathogens

orcid.org/0000-0003-2946-9497

Michael Malim

Editor-in-Chief

PLOS Pathogens

orcid.org/0000-0002-7699-2064
---

## [Editor Report · Acceptance letter]

Dear Dr. Barfod,

We are delighted to inform you that your manuscript, " Deposition of complement regulators on the surface of Plasmodium falciparum merozoites depends on the immune status of the host ," has been formally accepted for publication in PLOS Pathogens.

Best regards,

Sumita Bhaduri-McIntosh

Editor-in-Chief

PLOS Pathogens

orcid.org/0000-0003-2946-9497

Michael Malim

Editor-in-Chief

PLOS Pathogens

orcid.org/0000-0002-7699-2064